 

# A novel role for *Ets4* in axis specification and cell migration in the spider *Parasteatoda tepidariorum*

Matthias Pechmann[1]*, Matthew A Benton[1], Nathan J Kenny[2], Nico Posnien[3], Siegfried Roth[1]

[1]Developmental Biology, Institute of Zoology, University of Cologne, Cologne, Germany; [2]Life Sciences Department, The Natural History Museum, London, United Kingdom; [3]Department of Developmental Biology, University of Goettingen, Goettingen, Germany

**Abstract** Organizers play important roles during the embryonic development of many animals. The most famous example is the Spemann organizer that sets up embryonic axes in amphibian embryos. In spiders, a group of BMP secreting mesenchymal cells (the cumulus) functions as an organizer of the dorsoventral axis. Similar to experiments performed with the Spemann organizer, transplantation of the cumulus is able to induce a secondary axis in spiders. Despite the importance of this structure, it is unknown which factors are needed to activate cumulus specific gene expression. To address this question, we performed a transcriptomic analysis of early embryonic development in the spider *Parasteatoda tepidariorum.* Through this work, we found that the transcription factor *Pt-Ets4* is needed for cumulus integrity, dorsoventral patterning and for the activation of *Pt-hunchback* and *Pt-twist* expression. Furthermore, ectopic expression of *Pt-Ets4* is sufficient to induce cell delamination and migration by inducing a mesoderm-like cell fate.

**\*For correspondence:**
pechmanm@uni-koeln.de

**Competing interests:** The authors declare that no competing interests exist.

## Introduction

The self-regulatory capacities of vertebrate embryos were most famously demonstrated by Spemann and Mangold. They found that by grafting the dorsal-lip of an amphibian embryo (now known as the Spemann Organizer) to the ventral side of the host gastrula embryo it was possible to induce a secondary body axis (*Spemann and Mangold, 2001*; *De Robertis, 2009*; *Anderson and Stern, 2016*).

Intriguingly, spider embryos also have high self-regulatory capacities, even to the extent that twinning can occur spontaneously (*Napiórkowska et al., 2016*; *Oda and Akiyama-Oda, 2008*). During spider embryogenesis a group of migratory cells (the cumulus) is needed to break the radial symmetry of the early embryo and to induce the dorsoventral body axis (*Oda and Akiyama-Oda, 2008*; *Akiyama-Oda and Oda, 2003*, *2006*; *McGregor et al., 2008*; *Hilbrant et al., 2012*; *Mittmann and Wolff, 2012*; *Schwager et al., 2015*). Similar to the vertebrate experiments, Holm showed that transplanting cumulus material was able to induce a secondary axis in spider embryos (*Holm, 1952*). Modern work has shown that the cumulus signals via BMP signaling (again, similar to vertebrates). The mesenchymal cumulus cells are the source of the BMP receptor ligand Decapentaplegic (*Akiyama-Oda and Oda, 2006*). Interfering with the BMP signaling pathway by gene knockdown results in the loss of dorsal tissue identity, which in turn leads to completely radially-symmetric and ventralized embryos (*Akiyama-Oda and Oda, 2006*). The cumulus forms in the center of the so-called germ-disc (the embryonic pole of the embryo) and migrates underneath the ectoderm towards the rim of the disc. Arrival of the cumulus at the rim induces the opening of the germ-disc (*Oda and Akiyama-Oda, 2008*; *Akiyama-Oda and Oda, 2003*, *2006*; *McGregor et al., 2008*;

**eLife digest** At the earliest stages of animal development, embryos consisting of only a handful of cells must figure out where each part of their body will come from. The first step in this process is to determine what will be their head versus their tail, and what will be their front versus their back. Many animals use specialized groups of cells, called "organizers", to make this decision. This occurs in backboned animals – including humans – and also in distantly related animals such as spiders.

In spiders, the developing embryo must form an organizer called the "cumulus" or the spiderling will not develop correctly. In order to form and maintain the cumulus, various genes must be turned on in a carefully controlled order in exactly the right cells. Pechmann et al. have now discovered the role of a previously unknown gene (called *Pt-Ets4*) that marks the spot where the cumulus forms. This gene is required for cumulus maintenance and it also helps to activate a number of other cumulus-specific genes. When this gene is disrupted, the spider embryo does not properly differentiate its front from its back.

The findings presented by Pechmann et al. add to a growing foundation of studies aiming to understand how genes 'talk' to one another and organize embryos as they develop. In years to come, the unraveling of these gene pathways, where genes sequentially turn other genes on and off, will allow us to more fully understand how a single cell can grow into a complete adult animal.

*Hilbrant et al., 2012*; *Mittmann and Wolff, 2012*; *Schwager et al., 2015*). Cumulus migration is dependent on the Hh-signaling pathway (*Akiyama-Oda and Oda, 2010*) and it was shown that the knockdown of components of this signaling pathway results in cumulus migration defects and in the ectopic opening of the germ-disc (*Akiyama-Oda and Oda, 2010*).

How the cumulus is specified and forms is still under debate. During the formation of the germ-disc a small cluster of cells ingress and form an indentation where the future center of the fully formed germ-disc will be located. This cluster of cells appears as a visible spot and is called the primary thickening (*Akiyama-Oda and Oda, 2003*; *Hilbrant et al., 2012*). However, it is not clear whether all or only a subset of the cells of the primary thickening give rise to the cumulus, or if cumulus cells arise from subsequent cell invagination at the site of the primary thickening (*Oda and Akiyama-Oda, 2008*; *Akiyama-Oda and Oda, 2003*). Cell tracing (*Holm, 1952*; *Edgar et al., 2015*), as well as the expression of the endodermal marker *forkhead* (*Oda et al., 2007*) within the primary thickening/cumulus cells led to the suggestion that the primary thickening/cumulus cells are central endodermal cells (*Hilbrant et al., 2012*; *Oda et al., 2007*). However, these studies could not completely rule out that the labeled cumulus cells develop into cells of the visceral mesoderm (*Edgar et al., 2015*).

During the last 15 years, research focused on candidate genes known to be involved in development in *Drosophila melanogaster* has revealed several aspects of how spider embryos pattern their main body axis. However, there are many open questions regarding the early regulation of cumulus specific gene expression, cumulus establishment and maintenance.

To overcome the limitations of the candidate gene approach, we have carried out transcriptome sequencing of carefully staged embryos to find new genes involved in cumulus and axial patterning in the spider *Parasteatoda tepidariorum*. From this work, we have identified the transcription factor *Pt-Ets4* as a new gene expressed during early development and have found it to be expressed exclusively within the central primary thickening and the cells of the migrating cumulus. Our combined genetic and cellular analyses show that Pt-Ets4 is needed for the integrity of the cumulus. We found that the knockdown of this gene leads to embryos that show axis patterning defects reminiscent of BMP knockdown phenotypes, suggesting that an intact cumulus is needed to induce the formation of the bilaterally symmetric spider embryo. Importantly, Pt-Ets4 is necessary and sufficient for driving the early expression of *twist* (a gene involved in gastrulation and mesoderm formation in *Drosophila*) and *hunchback,* and the ectopic expression of Pt-Ets4 is sufficient to induce cell delamination.

## Results

The formation of the germ-disc is one of the most important events during spider embryogenesis. While a regular blastoderm (with no visible axial polarity) is present at stage 2, the germ-disc condenses during stage 3 of embryonic development (*Figure 1A and B*, for detailed description see [*Pechmann, 2016*]). This event leads to the establishment of the anterior/posterior body axis (anterior: rim of the disc; posterior: center of the disc).

To find new genes that are involved in the process of axis specification we sequenced the embryonic transcriptomes of stage 1, stage 2 and stage 3 embryos (*Figure 1—figure supplement 1*) and searched for genes showing a similar expression profile as *Pt-decapentaplegic* (*Pt-dpp*), *Pt-hedgehog* (*Pt-hh*) and *Pt-patched* (*Pt-ptc*) (representing genes that show early transcription and are required for axis formation or cumulus migration, respectively (*Akiyama-Oda and Oda, 2006*, *Akiyama-Oda and Oda, 2010*); *Figure 1—figure supplement 1*). One of the candidates was an *Ets*-like gene with high similarity to *Drosophila melanogaster Ets4/Ets98B* (see *Figure 1—figure supplement 2*), henceforth called *Parasteatoda tepidariorum Ets4* (*Pt-Ets4*).

### *Pt-Ets4* is expressed in the migrating cumulus

Prior to germ-disc condensation, *Pt-Ets4* is expressed within the cluster of cells that will form the center of the future germ-disc at early stage 3 (*Figure 1A*). Expression persists throughout germ-

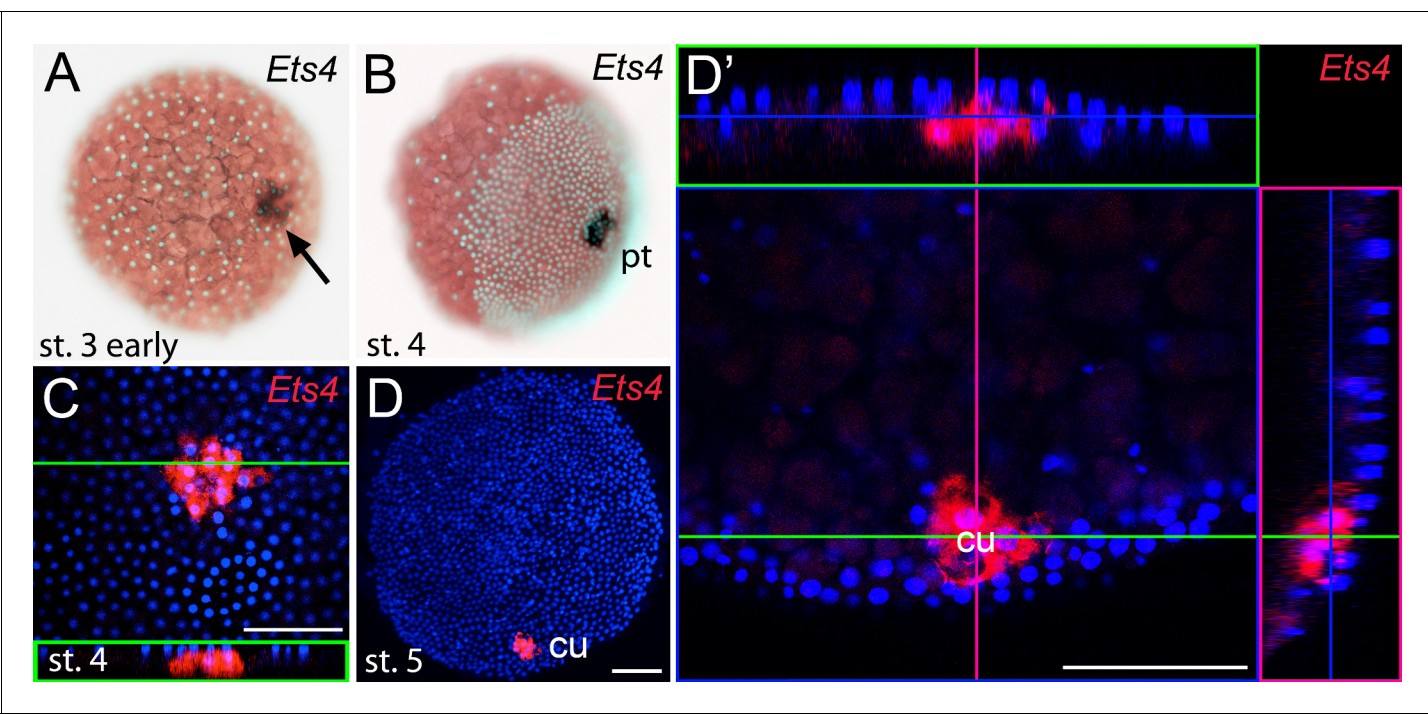

**Figure 1.** Early embryonic expression of *Pt-Ets4*. (A and B) *Pt-Ets4* is expressed within the cluster of cells (arrow in A) that will develop into the central primary thickening (pt) (B). (C–D') Confocal scans (single optical slices (C and D'); maximum intensity projection (D)) of embryos stained for *Pt-Ets4* (FastRed stain; red) and nuclei (DAPI; blue). *Pt-Ets4* is expressed within the primary thickening and in the migrating cumulus (cu), which are covered by the surface epithelium of the germ-disc. Orthogonal views are boxed in green and magenta. The same embryo is depicted in D and D'. Scale bar is 100 μm.

The following figure supplements are available for figure 1:

**Figure supplement 1.** Expression profile analysis.

**Figure supplement 2.** Phylogenetic analysis.

**Figure supplement 3.** Extended expression analysis of *Pt-Ets4*.

disc formation and, at stage 4, *Pt-Ets4* is strongly and exclusively expressed within the central cluster of cells (the so-called primary thickening, *Figure 1B*) that has delaminated during germ-disc formation (*Figure 1C*). During stage 5, the cumulus starts to migrate from the center of the germ-disc to its periphery (*Oda and Akiyama-Oda, 2008*; *Akiyama-Oda and Oda, 2003*, *2006*; *McGregor et al., 2008*; *Hilbrant et al., 2012*; *Mittmann and Wolff, 2012*; *Schwager et al., 2015*). At this stage *Pt-Ets4* is strongly and exclusively expressed in the migrating cumulus cells (*Figure 1D and D'*).

We were not able to detect *Pt-Ets4* transcripts in ovaries and early stage 1 embryos via RNA *in situ* hybridization (*Figure 1—figure supplement 3A and B*). In addition, our sequencing data shows that *Pt-Ets4* is only expressed at a very low level during early stage 1, but is mildly up-regulated at late stage 2 and strongly up-regulated at early stage 3 (*Figure 1—figure supplement 1*). From this we conclude that *Pt-Ets4* transcripts are not maternally provided.

## *Pt-Ets4* is necessary for the integrity of the cumulus

Time-lapse imaging and cross-sectioning revealed that the knockdown of *Pt-Ets4* neither affected formation of the germ-disc nor of the primary thickening/cumulus (*Video 1B*; middle column in *Figure 2*, *Figure 3A and B*). However, during stage 5, cumulus integrity was affected in *Pt-Ets4* knockdown embryos (*Video 1B*, middle column in *Figure 2*). While in control embryos the cumulus migrated towards the rim (*Video 1A*, left column in *Figure 2*), the cumulus of *Pt-Ets4* RNAi embryos remained at the center of the germ-disc until early stage 5 and disappeared soon after gastrulation was initiated at the center and at the rim of the disc (*Video 1B* (15h onwards), middle column in *Figure 2*). Analysis of mid-stage 5 *Pt-Ets4* RNAi embryos for the expression of the cumulus marker *Pt-fascin* (*Akiyama-Oda and Oda, 2010*) revealed that although the cells of the cumulus were still in the center of the germ-disc, they appeared to be more loosely organized (*Figure 3C and D*).

In *Pt-Ets4* knockdown embryos the radial symmetry of the germ-disc was not broken (as shown by the formation of a tube-like germ band in *Video 1B* (30h onwards), and the middle column in *Figure 2*), presumably due to the loss of the cumulus. To investigate this phenotype in more detail, we knocked down another gene that also results in defects in radial symmetry breaking, *Pt-ptc*. In *Pt-ptc* knockdown embryos, cumulus migration is lost but the cumulus itself is otherwise unaffected (*Akiyama-Oda and Oda, 2010*). After pRNAi with *Pt-ptc*, the cumulus stays in the center of the germ-disc (*Video 1C*; right column in *Figure 2*) and BMP signaling is ectopically activated (as shown via antibody staining against the phosphorylated form of mothers against dpp (pMAD) (*Akiyama-Oda and Oda, 2010*)). As a result, the germ-disc ectopically opened at the center and the dorsal field was induced at the posterior pole of the *Pt-ptc* RNAi embryos (right column in *Figure 2*). This ectopic induction of the dorsal field in the center of the germ-disc was never observed in the *Pt-Ets4* RNAi embryos. To test if the disappearing cumulus was at least partially able to activate BMP signaling in the germ-disc of *Pt-Ets4* RNAi embryos, we performed a pMAD antibody staining in both control and *Pt-Ets4* RNAi embryos. In control embryos the cumulus reached the periphery of the germ-disc at late stage 5 and a strong pMAD staining was visible in the overlying ectodermal cells (*Figure 3E*, *Figure 3—figure supplement 1*). At this stage, the anterior marker *Pt-orthodenticle* (*Pt-otd*) was expressed in a ring, which had a width of 3–5 cells (*Akiyama-Oda and Oda, 2003*; *Pechmann et al., 2009*; *Akiyama-Oda and Oda, 2016*) (*Figure 3E*). In *Pt-Ets4* knockdown embryos *Pt-otd* expression was unaffected but nuclear pMAD was not detectable (*Figure 3F*). This lack of BMP signaling explains why *Pt-Ets4* RNAi embryos do not induce the dorsal field and stay radially symmetric. Indeed, the

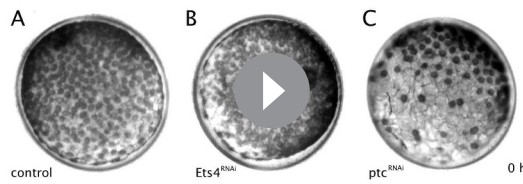

**Video 1.** Knockdown of *Pt-Ets4* and *Pt-ptc*. Live imaging of a control (A), a *Pt-Ets4* RNAi (B) and a *Pt-ptc* RNAi (C) embryo under transmitted light conditions. The video starts at stage 3 and ends at stage 9 of embryonic development. Cumulus migration and normal germ-band formation is visible in the control embryo (A). The cells of the cumulus disperse in the *Pt-Ets4* knockdown embryo (B). The ventralized *Pt-Ets4* RNAi embryo stays radially symmetric and posterior tube formation is initiated (30 hr onwards). The cumulus of the *Pt-ptc* RNAi embryo does not migrate (C). The germ-disc opens up at the central position and the radially symmetric embryo overgrows the yolk and anterior tube formation is initiated (48 hr onwards).

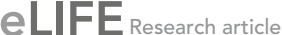

**Figure 2.** *Pt-Ets4* is required for cumulus integrity. Stills from the embryos shown in *Video 1*. The cumulus (asterisk) migrates in the control, disappears in the *Pt-Ets4* RNAi and stays in the center of the germ-disc in *Pt-ptc* RNAi embryo. Ectopic, central opening (induction of the dorsal field) of the germ-disc is depicted via the dotted line (*Pt-ptc* RNAi st. 6). Posterior (P) and anterior (A) tube formation in *Pt-Ets4* and *Pt-ptc* knockdown embryos is also indicated.

*Figure 2 continued on next page*

*Figure 2 continued*

The following figure supplement is available for figure 2:

**Figure supplement 1.** Knockdown efficiency after RNAi with *Pt-Ets4*.

knockdown embryos were completely ventralized; the expression of the ventral marker *Pt-short-gastrulation* (*Pt-sog*) was uniform around the embryonic circumference and the segmental marker *Pt-engrailed* (*Pt-en*) was expressed in symmetric rings demonstrating the radial symmetry of the embryo (*Figure 3G–L*). During later development the embryonic tissue either grew completely over the yolk (*Figure 3J–L*), or a tube like structure elongated at the posterior of the embryo (*Video 1B*, middle column *Figure 2*). This was in contrast to *Pt-ptc* RNAi embryos, where the germ-disc opened up centrally and a tube like structure formed at the anterior of the embryo (right column *Figure 2*; *Video 1C*).

The *Pt-Ets4* knockdown phenotype is rather similar to that of *Pt-dpp*, although cumulus migration/integrity is not affected in *Pt-dpp* RNAi embryos (*Akiyama-Oda and Oda, 2006*). This observation, plus the early and strong expression of *Pt-Ets4* and the fact that BMP signaling disappeared upon *Pt-Ets4* knockdown, led us to hypothesize that *Pt-Ets4* functions as an activator of *Pt-dpp* transcription within the cells of the cumulus. However, there was no obvious difference in the expression of *Pt-dpp* in stage 4 control and *Pt-Ets4* RNAi embryos (*Figure 4A and E*). Vice versa, *Pt-dpp* appears not to be regulating the expression of *Pt-Ets4* (*Figure 4—figure supplement 1A*). In addition, Hh signaling seems not to be involved in the regulation of *Pt-Ets4* expression and *Pt-Ets4* appears not to regulate the expression of *Pt-ptc* within the primary thickening (*Figure 4—figure supplement 1B and C*).

In order to determine what other genes *Pt-Ets4* may regulate, we studied the expression of genes normally expressed in the primary thickening in *Pt-Ets4* RNAi embryos. We found that *Pt-forkhead* (*Pt-fkh*) expression was unaffected (*Figure 4B and F*), while the cumulus marker *Pt-fascin* was slightly down regulated, and *Pt-hunchback* (*Pt-hb*) was strongly down regulated in *Pt-Ets4* knockdown embryos (*Figure 4C,D,G and H*).

## Ectopic expression of *Pt-Ets4* induces cell migration

As *Pt-Ets4* is strongly expressed in the cumulus and is required for cumulus integrity, we wondered how ectopic expression of *Pt-Ets4* would affect cell behavior. To generate small cell clones ectopically expressing Pt-Ets4 within the germ-disc, we micro-injected late stage 1 embryos (via single cell/blastomere injections [*Hilbrant et al., 2012*; *Kanayama et al., 2010*]) with in vitro synthesized capped mRNA coding for an EGFP-Pt-Ets4 fusion protein (see *Figure 5—figure supplement 1*).

Our first observation was that the EGFP-Pt-Ets4 fusion protein localizes to the nuclei of the injected cells, suggesting that the nuclear-localization-signal of Pt-Ets4 is functioning normally (*Figure 5A*). EGFP-Pt-Ets4 marked cell clones resembled wild-type until stage 4. As soon as a dense germ-disc formed, however, the cell clones expressing EGFP-Pt-Ets4 seemed to move beneath the germ-disc and the EGFP signal became occluded by the opaque cells of the germ-disc (*Figure 5B*). We have never seen such behavior following injection of EGFP-NLS constructs alone (*Figure 5F–F'''*), indicating that the change in cell behavior is due to the ectopic expression of *Pt-Ets4*.

To further visualize the process of cell clone delamination, we marked cell membranes by co-injecting capped mRNA coding for EGFP-Pt-Ets4 together with capped mRNA coding for lynGFP (*Köster and Fraser, 2001*). In control embryos (ectopic expression of lynGFP alone), lynGFP strongly marked the cell outlines and cell clones stayed at the surface epithelium of the germ-disc (*Video 2A*, *Figure 5C–C'''*). Regardless of the position and the shape of the cell clone, cells expressing lynGFP/EGFP-Pt-Ets4 constricted and delaminated shortly after the formation of the germ-disc (*Video 2B–D*, *Figure 5D–D'''*). The detection of EGFP-Pt-Ets4 cell clones in fixed embryos using an antibody against EGFP confirmed that the labeled cells were below the epithelium of the germ-disc (*Figure 5E and E'*).

The nuclear signal of the EGFP-Pt-Ets4 fusion construct (and the membrane signal of the lynGFP) is hardly visible after the delamination process. Therefore, we co-injected capped mRNA for EGFP-

**Figure 3.** Cumulus integrity and signaling is affected in *Pt-Ets4* knockdown embryos. (**A and B**) Cross-section through the central cumulus of ubiquitously stained (via *Pt-arm* RNA *in situ* hybridization) control (**A**) and *Pt-Ets4* RNAi (**B**) embryos. (**C and D**) control and *Pt-Ets4* RNAi embryos stained for the cumulus marker *Pt-fascin*. Cells of the cumulus are dispersing in the *Pt-Ets4* knockdown embryo (**D**). (**E and F**) Single color double stain of anterior *Pt-otd* expression (anterior ring) and nuclear localized pMAD in the cells overlaying the cumulus. pMAD signal is absent in *Pt-Ets4* RNAi embryos (**F**). *In situ* hybridization for the ventral fate marker *Pt-sog* (**G, G', J and J'**) or the segmental marker *Pt-en* (**H, I, K and L**) in control (**G–I**) and *Pt-Ets4* knockdown embryos (**J–L**). The same embryos in fluorescence vs. bright field channel are shown in G and G' as well as in J and J'. Nuclear stain (DAPI)/bright field overlay is shown in A-F, H, I, K and L. Flat mounted embryos in C-F. Lateral-ventral view (**G, G', J–K**), lateral view (**H**), ventral view (**I**). Abbreviations: ch: cheliceral segment; L1-L4: walking leg bearing segments 1–4; O1: opisthosomal segment 1.

The following figure supplement is available for figure 3:

**Figure supplement 1.** BMP pathway activity in wt embryos.

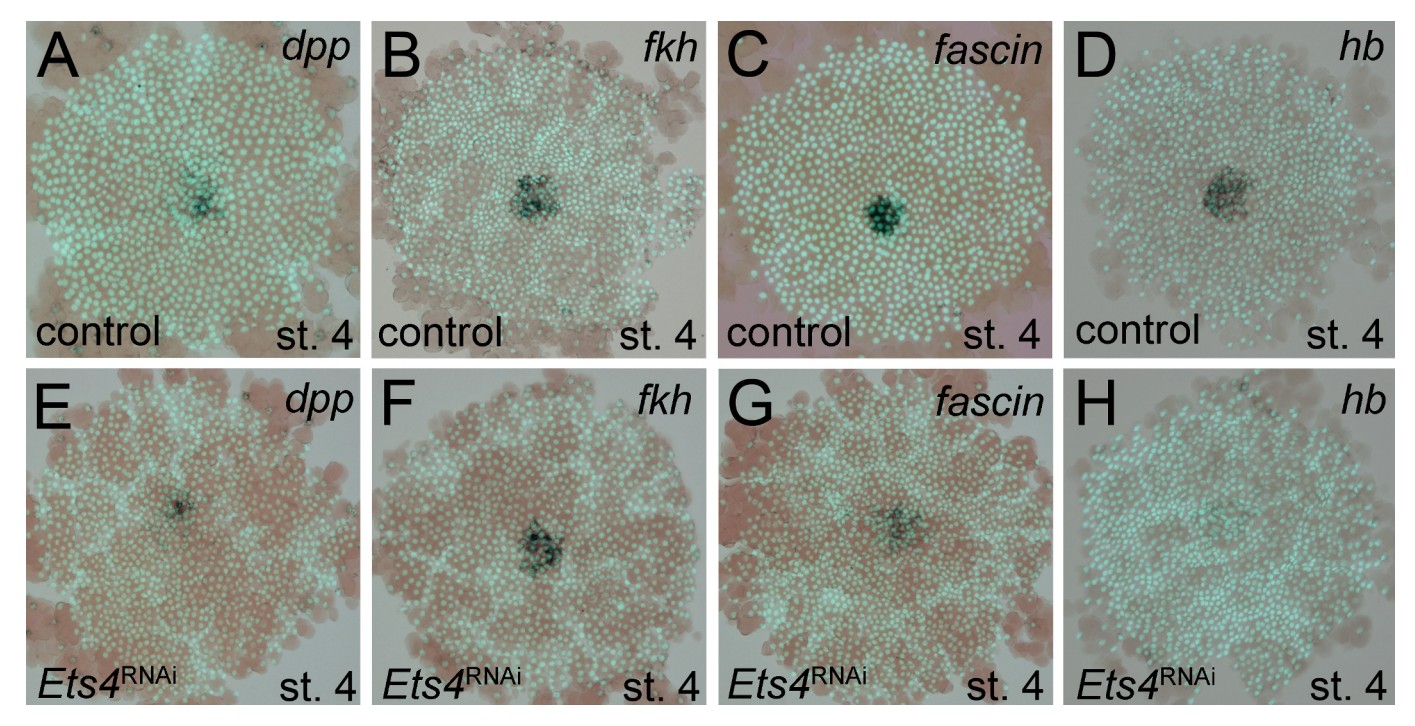

**Figure 4.** Analysis of 'cumulus marker' genes. Flat mount preparation of *in situ* stained stage 4 embryos. While *Pt-dpp* and *Pt-fkh* expression is unaffected (compare A to E and B to F), *Pt-fascin* expression is slightly (compare C to G) and *Pt-hb* expression is strongly down regulated in *Pt-Ets4* RNAi embryos (compare D to H).

The following figure supplement is available for figure 4:

**Figure supplement 1.** Regulation of *Pt-Ets4* and *Pt-ptc.*

Pt-Ets4 with capped mRNA for nuclear localized EGFP (EGFP-NLS, see *Figure 5—figure supplement 1*), a construct that we have found to produce a very bright and persistent fluorescent signal. This experiment resulted in a strong nuclear localized EGFP signal within the marked cell clone, which we used to perform time-lapse imaging. While in the control embryos (injected with EGFP-NLS alone) the marked cell clones persisted at the surface of the germ-disc (*Video 3A*, *Figure 5F and F'*), the cell clone expressing both EGFP-NLS and EGFP-Pt-Ets4 delaminated shortly after the formation of the germ-disc (*Video 3B*, *Figure 5G and G'*). As previously shown (*Kanayama et al., 2010*, *Kanayama et al., 2011*), germ-disc cells continue to divide and undergo convergent extension during the formation of the germ-band, which causes cell clones to become thin and elongated as seen in our control EGFP-NLS clones (*Video 3A*, *Figure 5F'' and F'''*). In contrast to this, cell clones expressing both EGFP-NLS and EGFP-Pt-Ets4 stopped dividing as soon they delaminated. In addition, when the cumulus started to migrate, the delaminated cells of the EGFP-NLS/EGFP-Pt-Ets4 marked cell clone lost contact with each other and spread out underneath the germ-disc/germ-band epithelium (*Video 3B*, *Figure 5G'' and G'''*). This observation was consistent and reproducible in multiple analyzed embryos (*Video 4*).

These results demonstrate that *Ets4* expression is sufficient to induce cell delamination and migration in embryos of *P. tepidariorum*.

## Ectopic expression of Pt-Ets4 induces mesoderm-like fate

As already mentioned, Pt-Ets4 seems to have no influence on the early expression of *Pt-dpp* itself (*Figure 4A and E*). Furthermore, we were not able to detect *Pt-dpp* transcripts in cells ectopically expressing Pt-Ets4 (*Figure 6A and A'*). In contrast to *Pt-dpp,* we did find that *Pt-Ets4* is regulating the expression of *Pt-hb*. While *Pt-hb* expression was nearly absent in *Pt-Ets4* RNAi embryos

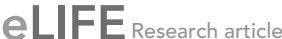

**Figure 5.** Ectopic expression of *Pt-Ets4* causes the delamination and migration of cells. (**A and A'**) A stage 4 embryo in which a cell clone has been marked via the ectopic expression of an EGFP-Pt-Ets4 fusion construct. The fusion protein localizes to the nuclei. (**B and B'**) The cell clone has delaminated four hours later. As the overlaying epithelium is highly light-scattering, the nuclear EGFP signal is no longer visible. The inset shows the magnification of the boxed region in A and B. (**C and D**) Stills from *Video 2A and D* (magnifications of the lynGFP positive regions). Cells expressing lynGFP alone (**C–C''**) stay at the surface epithelium of the germ-disc. Cells expressing lynGFP/EGFP-Pt-Ets4 (**D–D''**) constrict and delaminate. (**E**) EGFP antibody staining (maximum intensity projection is shown in E and the orthogonal view is shown in E') of a fixed embryo ectopically expressing EGFP-Pt-Ets4. (**F–F'''**) Stills from *Video 3A* (control). A cell clone marked via the ectopic expression of nuclear localized EGFP (EGFP-NLS) marks the ectoderm during germ-band formation. (**G–G'''**) Stills from *Video 3B* (ectopic expression of *Pt-Ets4*). A cell clone ectopically expressing EGFP-Pt-Ets4 in combination with EGFP-NLS delaminates after germ-disc formation (**G'**). The cells of this cell clone start to disperse during later stages of development (**G'' and G'''**). Scale bar is 100 μm in E.

The following figure supplement is available for figure 5:

**Figure supplement 1.** Constructs.

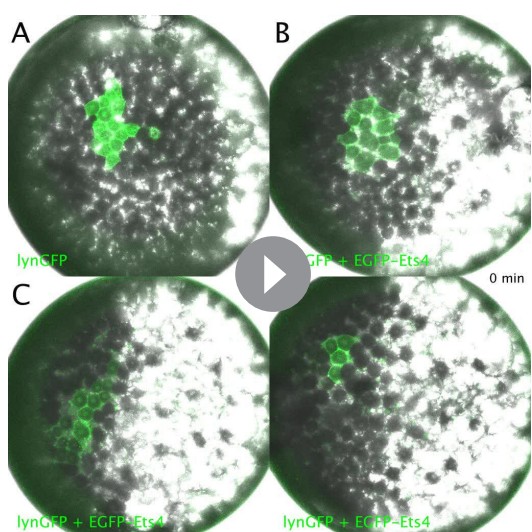

**Video 2.** Ectopic expression of Pt-Ets4 causes the delamination of cells. Ectopic expression of lynGFP (A) and lynGFP in combination with EGFP-Ets4 (B-D). The lynGFP positive cell clone (A) stays in the ectodermal cell layer of the germ-disc. Regardless of the shape of the cell clones, Pt-Ets4 positive cells apically constrict and delaminate (B-D).

(*Figure 4H*), *Pt-hb* transcripts were present in the ectopically *Pt-Ets4* positive cell clone (*Figure 6B and B'*).

The behavior of the Pt-Ets4 positive cells is reminiscent of migrating gastrulating cells that invade the germ-disc from the center and from the rim of the germ-disc (*Mittmann and Wolff, 2012*; *Kanayama et al., 2011*; *Yamazaki et al., 2005*) (see *Video 1A*). As Pt-Ets4 activates the expression of *Pt-hb* (a gene that is known to be expressed in mesodermal cells in diverse animals [*Schwager et al., 2009*; *Franke and Mayer, 2015*; *Kerner et al., 2006*]) we wondered if *Pt-Ets4* misexpression is inducing a mesoderm-like cell fate. For this reason, we tested whether the ectopic expression of *Pt-Ets4* is also inducing the expression of the key mesodermal marker *Pt-twist* (*Pt-twi*). Indeed, *Pt-twi* was detectable within the cell clone that ectopically expressed Pt-Ets4 (*Figure 6C and C'*, compare to controls in *Figure 6—figure supplement 1*). Interestingly, in the stage 4 embryos ectopically expressing Pt-Ets4, we could not only detect *Pt-twi* transcripts within the ectopic Pt-Ets4 expressing cell clone, but also within the central primary thickening (*Figure 6C'*). This comes as a surprise, as it was reported that *Pt-twi* expression is not initiated before the end of stage 5 (*Yamazaki et al., 2005*). For this reason, we reanalyzed the full expression series of *Pt-twi* in wild-type embryos and we were able to confirm that *Pt-twi* is expressed in the developing primary thickening of stage 3 and 4 embryos (*Figure 6—figure supplement 2A–C*). Finally, we confirmed the regulation of *Pt-twi* via *Pt-Ets4* by analyzing the expression of *Pt-twi* in *Pt-Ets4* knockdown embryos. *Pt-twi* transcripts were no longer detectable in the primary thickening of stage 4 *Pt-Ets4* RNAi embryos (*Figure 6D and E*).

However, late segmental mesoderm specification was unaffected, as the expression of *Pt-twi*

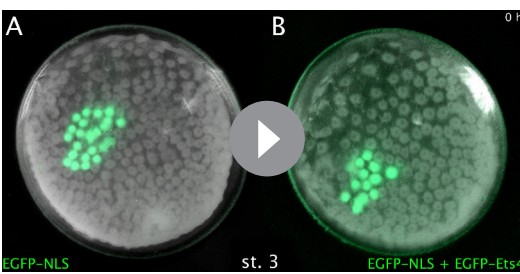

**Video 3.** Ectopic expression of Pt-Ets4 causes the delamination and migration of cells. Ectopic expression of EGFP-NLS (A) and EGFP-NLS in combination with EGFP-Ets4 (B). The cell clone positive for EGFP-NLS (A) stays in the ectodermal cell layer of the germ-disc. Cells further divide and form a long and thin cell clone (via convergent extension [*Kanayama et al., 2011*]) at stage 8 of embryonic development. The Pt-Ets4 positive cell clone delaminates at stage 4 (B). Cells stop dividing and start to disperse as soon as the cells of the cumulus start to migrate (st. 5).

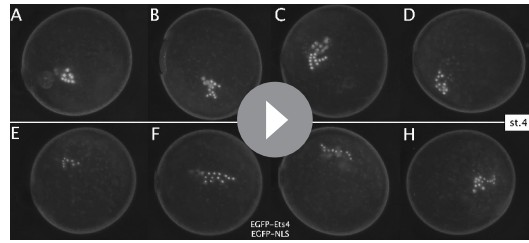

**Video 4.** Ectopic expression of EGFP-NLS and EGFP-Pt-Ets4 in multiple embryos. Ectopic expression of EGFP-NLS (A-D) and EGFP-NLS in combination with EGFP-Ets4 (E-H). EGFP-NLS (A-D) and ectopic clones expressing Pt-Ets4 (E-H) have equal positions within the germ-disc. While all of the control cell clones (A-D) form long and thin stretched cell clones at stage 8 of embryonic development, the Pt-Ets4 positive cells clones (E-H) delaminate and the cells disperse from stage 5 onwards. Only the EGFP channel is shown for all embryos.

**Figure 6.** *Pt-Ets4* regulates *Pt-hb* and *Pt-twi* expression within the primary thickening. Live stage 4 embryos in which a cell clone is marked via the ectopic-expression of EGFP-Ets4 are depicted in A-C. The same embryos have been fixed and analyzed for their expression of *Pt-dpp* (**A'**), *Pt-hb* (**B'**) and *Pt-twi* (**C'**), respectively. Expression of *Pt-twi* in a control (**D**) and a *Pt-Ets4* knockdown embryo (**E**). A', B', C', D and E are false-color overlays of *in situ* hybridization images.

*Figure 6 continued on next page*

*Figure 6 continued*

The following figure supplements are available for figure 6:

**Figure supplement 1.** Controls for the ectopic expression of EGFP-Ets4.

**Figure supplement 2.** Expression of *Pt-twi* in wt and *Pt-Ets4* RNAi embryos.

was unchanged in stage 7 *Pt-Ets4* knockdown embryos (*Figure 6—figure supplement 2H–I'*). This confirms that the formation and migration of the cumulus and later gastrulation events from central and peripheral parts of the germ-disc are two independent processes (*Hilbrant et al., 2012*; *Oda et al., 2007*).

Overall, we suggest that the activation of *Pt-twi* and *Pt-hb* by Pt-Ets4 in cell clones initiated a mesoderm-like cell fate that led to the migratory behavior of the ectopic Pt-Ets4 positive cells.

## Discussion

Ets proteins belong to a highly conserved family of transcription factors (*Laudet et al., 1999*) that play roles in a variety of different cellular processes (e.g. growth, migration, differentiation) and it has been shown that these factors can act as activators or repressors of transcription (*Oikawa and Yamada, 2003*; *Sharrocks et al., 1997*; *Sharrocks, 2001*; *Hollenhorst et al., 2011*). In *D. melanogaster*, *Ets4* (also known as *Ets98B*) is expressed in the oocyte nucleus and the primordial germ cells (PGC) (*Chen et al., 1992*; *Hsouna et al., 2004*) and there is evidence that *Dm-Ets4* is involved in the migration of the PGCs (*Hsouna et al., 2003*). Furthermore, the mammalian homolog of *Ets4*, *Pdef*, is down regulated in invasive and migratory breast tumor cells (*Feldman et al., 2003*). Together, these reports suggest that Ets4 regulates migratory cell behavior in different organisms.

Here we show that in the spider *P. tepidariorum*, *Ets4* is strongly expressed in the migrating cumulus and that *Pt-Ets4* is needed for the integrity of the cumulus, a group of cells, which need to migrate together in order to function normally. We also show that the ectopic expression of *Pt-Ets4* is able to induce cell delamination and cell migration within the ectodermal germ-disc cells (a process that is known as epithelial-to-mesenchymal transition [*Thiery et al., 2009*; *Gilmour et al., 2017*]). Taken together, these findings suggest that *Pt-Ets4* likely plays an important role in the migratory behavior of the cumulus.

It has been suggested that the cumulus cells of many higher spiders are specified during early embryogenesis and are not secondarily induced after the formation of the germ-disc (*Edgar et al., 2015*). *Pt-Ets4* marks the cells of the developing primary thickening (st. 3 and 4) and of the migrating cumulus. Therefore, our results indicate that in *P. tepidariorum* as well, cumulus cells are induced before germ-disc formation is complete.

As the primary thickening forms in the absence of *Pt-Ets4*, *Pt-Ets4* cannot be the only factor that is required for the specification of the cells that will develop into the primary thickening (and later into the cumulus). This is supported by our finding that *Pt-Ets4* is sufficient to induce the delamination and migration of ectodermal cells, but it is not sufficient to induce the formation of a fully functional ectopic cumulus. So far, there have been no reports of any gene knockdown that completely inhibits the formation of the primary thickening. The cumulus is characterized by the co-expression of multiple genes, including *Pt-dpp*. As shown by our analyses, *Pt-Ets4* seems not to be involved in the regulation of *Pt-dpp* itself, as well as several other cumulus marker genes. Therefore, important factors of the cumulus are missing in the ectopic *Pt-Ets4* expression cells. Taken together, our findings indicate that the formation of the primary thickening is a process that is buffered via the input of several genes or other unknown mechanisms/factors.

Our results also show that an intact cumulus is needed to open up the germ-disc and to initiate the formation of a bilaterally symmetric spider embryo. In the germ-disc of *P. tepidariorum* embryos, BMP signaling is active from early stage 5 (*Figure 3—figure supplement 1*). However, the germ-disc does not open up before the arrival of the cumulus at the rim of the disc at the end of stage 5 even though the putative receptors are ubiquitously expressed. Therefore, the establishment of the dorsal field is a precisely timed process, and it is possible that in *Pt-Ets4* knockdown embryos the

cells of the cumulus disperse too early, and are no longer able to induce the opening of the germ-disc at the end of stage 5. An alternative, although not mutually exclusive, explanation involves the fact that the cumulus may signal via cytonemes (*Akiyama-Oda and Oda, 2003*; *Hilbrant et al., 2012*). These structures could be affected in *Pt-Ets4* RNAi embryos, as the cumulus cells lose contact with each other. Lastly, *Pt-Ets4* could be involved in regulating the BMP pathway activity at a different level (e.g. protein-protein interactions, protein modifications) or the knockdown of *Pt-Ets4* could result in a fate change of the cumulus cells before cumulus migration is initiated. Investigation of each of these hypotheses will require the establishment of advanced techniques to study cell migration, cell microstructure, and protein interactions in *P. tepidariorum*.

Via both our loss- and gain-of-function experiments, we found that *Pt-Ets4* is involved in the activation of at least two genes that are expressed within the primary thickening of stage 4 embryos. These genes are *Pt-hb* and *Pt-twi*. For *Pt-hb*, it was already shown that this gene is strongly expressed within the primary thickening of stage 4 embryos (*Schwager et al., 2009*). However, it was reported that *Pt-twi* expression is not detectable before late stage 5 (*Yamazaki et al., 2005*), while we found that *Pt-twi* is expressed in the primary thickening already at stage 3 and 4. This is in agreement with our sequencing data, which shows that *Pt-twi* is strongly up regulated from stage 2 to stage 3 (*Figure 1—figure supplement 1*). This result indicates that the cumulus cells are not endodermal in nature (*Oda and Akiyama-Oda, 2008*; *Hilbrant et al., 2012*; *Oda et al., 2007*) but rather are mesendodermal. For this reason, the genetic composition and the eventual fate of the cumulus should be closely examined in future studies.

## Conclusions

The cumulus is a fascinating example of a migrating and signaling organizer. Thus far, cumulus-related dorsoventral patterning defects have been observed in spider embryos that either completely lack BMP signaling or are deficient for cumulus migration (*Akiyama-Oda and Oda, 2006*, *Akiyama-Oda and Oda, 2010*). Here, we show that the knockdown of the transcription factor *Pt-Ets4* generates a novel dorsoventral phenotype that is dependent on cumulus integrity. Our results show that formation of the bilaterally symmetric spider embryo is a precisely timed process that relies on the presence of an intact, migrating and signaling cumulus.

## Materials and methods

### Spider husbandry and embryology

*Parasteatoda tepidariorum* adults and embryos were obtained from our laboratory culture at the University of Cologne. Spiders were kept in plastic vials at room temperature and fed with *Drosophila melanogaster* and crickets (*Acheta domesticus* and *Gryllus bimaculatus*). Embryos were staged according to (*Mittmann and Wolff, 2012*).

### Gene cloning

PCR amplification and cloning of *Pt-Ets4* was performed using standard techniques. *Pt-twist* (AB167807.1), *Pt-hunchback* (FM956092.1), *Pt-engrailed* (AB125741.1), *Pt-fork-head* (AB096073.1), *Pt-armadillo* (AB120624.1), *Pt-orthodenticle* (AB096074.1), *Pt-short-gastrulation* (AB236147.1), *Pt-decapentaplegic* (AB096072.1) and *Pt-fascin* (AB433905.1) have been isolated previously.

### Identification and cloning of *Pt-Ets4* sequence

The transcriptomes of the embryonic stages 1–3 (early stage 1, late stage 2 and early stage 3; see *Figure 1—figure supplement 1*) were sequenced (HiSeq2000) at the Cologne Center for Genomics. The total RNA of three cocoons per stage was pooled and sequenced in each case.

The sequence reads (deposited to the Sequence Read Archive (http://www.ncbi.nlm.nih.gov/sra/ (RRID:SCR_004891), BioProject ID: PRJNA383558) were mapped to the AUGUSTUS gene predictions (RRID:SCR_008417) (https://i5k.nal.usda.gov/Parasteatoda_tepidariorum, *Schwager et al., 2017*) using Bowtie 2 (*Langmead and Salzberg, 2012*). New candidates were picked according to their expression profile. These new candidates (including *Pt-Ets4*; AUGUSTUS prediction: *aug3.g4238*) were up-regulated in a similar manner as *Pt-decapentaplegic*, *Pt-hedgehog* and *Pt-patched* (genes that show a defect in dorsoventral patterning or cumulus migration upon knockdown

(*Akiyama-Oda and Oda, 2006*, *Akiyama-Oda and Oda, 2010*) (see *Figure 1—figure supplement 1*). A 1094 bp fragment of *Pt-Ets4* was amplified using the primer Pt-g4238-Fw (5′-GTA CAC AGC ACC TTC TAT TAT GG-3′) and Pt-g4238-Rev (5′-CCT TCT TGT AAT ATT GGC GA-3′) in an initial PCR. For the production of dsRNA a T7 promoter sequence was added to the 5′ and 3′ end of the sequence by performing a nested PCR with the primer T7-Pt-g4238-Fw (5′-GTA ATA CGA CTC ACT ATA GGG CCA CAA AAG ATG GCC-3′) and T7-Pt-g4238-Rev (5′-GTA ATA CGA CTC ACT ATA GGG GAA CGG CTG AGT TTG-3′). This nested PCR yielded a 1046 bp fragment that was used for the initial knockdown of *Pt-Ets4*.

## RNAi

Double stranded RNA (dsRNA) was produced using the MEGAscript T7 Kit (ThermoFisher SCIENTIFIC).

Within one week, adult females of *Parasteatoda tepidariorum* were injected three to four times with 2 µl dsRNA solution (2–3 µg/µl). Water injections served as a control.

The knockdown of *Pt-Ets4* was performed several times ($n_{experiments} > 5$; $n_{injected\ females} > 24$) and always resulted in the same phenotype. For the statistical analysis (*Figure 2—figure supplement 1D–F*) two non-overlapping fragments, targeting the CDS and the 3′UTR of *Pt-Ets4,* were used (see *Figure 2—figure supplement 1C*). The coding sequence of *Pt-Ets4* was amplified using the primer Pt-CDS-Ets4-Fw (5′-GTA GTC TTG AAC TTC AGT TAT CAA AG-3′) and Pt-CDS-Ets4-Rev (5′-GGT TTA CTT CAA GAA CTG GAC-3′) and was cloned into the pCR4 vector (ThermoFisher SCIENTIFIC). The 3′UTR of *Pt-Ets4* was amplified using the primer Pt-3′Ets4-Fw (5′-CAC TAT GGT TTC AAA CAT CGA TTG-3′) and Pt-3′Ets4-Rev (5′-GTC ATA TCC CCT CTA TAG CTA AC-3′) and was cloned into the pCRII-Blunt vector (ThermoFisher SCIENTIFIC). For the production of dsRNA the T7 promoter sequence was added to both ends of the CDS and the 3′UTR fragment by using the primer T7-Pt-CDS-Ets4-Fw (5′-GTA ATA CGA CTC ACT ATA GGG GTA GTC TTG AAC TTC AGT TAT C-3′) and T7-Pt-CDS-Ets4-Rev (5′-GTA ATA CGA CTC ACT ATA GGG GTC TGA AGT AAT CTT CTG ATA G-3′) and T7-Pt-3′Ets4-Fw (5′-GTA ATA CGA CTC ACT ATA GGG CAC TAT GGT TTC AAA CAT CG-3′) and T7-Pt-3′Ets4-Rev (5′-GTA ATA CGA CTC ACT ATA GGG CCT AAA ACA CAG TTT TAG GAG-3′), respectively. We observed a similar knockdown efficiency for both fragments with the highest penetrance in the third and fourth cocoons (*Figure 2—figure supplement 1E and F*). As many embryos were able to recover from the knockdown of *Pt-Ets4* (one analyzed cocoon had a recovery rate of 71%; n = 66) during later stages of development (>stage 8, see *Figure 2—figure supplement 1G*), the number of affected embryos was analyzed during the embryonic stages 6 and 7.

A gene fragment of *Pt-decapentaplegic* was amplified using the primers Pt-dpp-Fw (5′-GTG ATC ATA ACA GGT TCC TGA CC-3′) and Pt-dpp-Rev (5′-GAC AAA GAA TCT TAA CGG CAA CC-3′). The resulting 1147 bp *Pt-dpp* fragment was cloned into pCRII-Blunt vector. dsRNA template was generated by using T7 and T7Sp6 primer. We injected three adult females of *P. tepidariorum* with *Pt-dpp* dsRNA and the knockdown resulted in the same phenotype as published (*Akiyama-Oda and Oda, 2006*). We used the *Pt-dpp* pRNAi embryos of a fourth cocoon (the development of 73 embryos of this cocoon were monitored under oil; one embryo died and 72 embryos showed a strong BMP signaling defect phenotype (*Akiyama-Oda and Oda, 2006*) during stages 6–8) to perform the *Pt-Ets4 in situ* staining shown in *Figure 4—figure supplement 1A*.

Two gene fragments of *Pt-patched* were amplified from a plasmid (containing a 2 kb fragment of *Pt-ptc*) using the primer T7-Pt-ptc-Fw1 (5′-GTA ATA CGA CTC ACT ATA GGG GGG TAG AAG ACG GCG G-3′) and T7-Pt-ptc-Rev1 (5′-GTA ATA CGA CTC ACT ATA GGG GAG ACT CTT AGC TA TAA TCT C-3′) and T7-Pt-ptc-Fw2 (5′-GTA ATA CGA CTC ACT ATA GGG GAG ATT ATA GCT AAA GAG TCT C-3′) and T7-Pt-ptc-Rev2 (5′-GTA ATA CGA CTC ACT ATA GGG GAT TTG TTT GTC GAC CAC C-3′). dsRNA of both *Pt-ptc* fragments were combined and injected into three adult *P. tepidariorum* females. The knockdown resulted in the same phenotype as published (*Akiyama-Oda and Oda, 2010*) (see right column in *Figure 2*, *Video 1*).

## Phylogenetic analysis

Amino acid sequences were obtained from FlyBase (RRID:SCR_006549) (*dos Santos et al., 2015*), WormBase (RRID:SCR_003098) (WormBase release Version: WS257), or translated from the *P. tepidariorum* AUGUSTUS predictions online (https://i5k.nal.usda.gov/Parasteatoda_tepidariorum). Amino

acid sequences were aligned using MUSCLE (RRID:SCR_011812) (*Edgar, 2004*), alignments were trimmed using TrimAl with the GappyOut setting (*Capella-Gutiérrez et al., 2009*), and maximum likelihood based phylogenies were constructed using PhyML at 'phylogeny.fr' (*Dereeper et al., 2008*). Full amino acid sequences were used for all genes except for *Pt-aug3.g5814.t1*, which is missing the N-terminus but still contains the ETS domain (as predicted online [*de Castro et al., 2006*]). Final phylogenies were generated with the WAG substitution model and 1000 bootstrap replicates (*Whelan and Goldman, 2001*). Phylogenetic analysis was also performed using the ETS domains alone, and while tree topology changed in some ways, the *Ets4* genes from *D. melanogaster* and *C. elegans* still branched together with strong support, and the gene we have named *Pt-Ets4* was the only *P. tepidariorum* gene branching together with this clade.

## Ectopic expression of Pt-Ets4 and EGFP

Experiments have been performed by injecting capped mRNA into late stage 1 embryos of *P. tepidariorum*. Embryonic microinjections were performed as described previously (*Pechmann, 2016*).

For the production of capped mRNA, the mMASSAGE mMACHINE Kit (T7 or Sp6, ThermoFisher SCIENTIFIC) was used. Capped mRNA was injected at a concentration of 2–3 µg/µl.

For the ectopic expression of *Pt-Ets4* an EGFP-Pt-Ets4-PolyA fusion construct was synthesized at Eurofins Genomics (see *Figure 5—figure supplement 1*; see *Supplementary file 1* for full sequence). For the production of capped mRNA, the construct contained a T7 and a Sp6 promoter at its 5' end and could be linearized via NotI, PstI or EcoRI restriction enzyme digest. In addition, the coding sequence of EGFP-Pt-Ets4 was flanked by the 5' and the 3' UTR of the *Xenopus* beta-globin gene (also used in *Tribolium* [*Benton et al., 2013*]). For the ectopic expression of NLS-EGFP, the *Pt-Ets4* sequence of the EGFP-Pt-Ets4-PolyA construct was removed (via BglII, SalI double digest) and replaced by the sequence MAKIPPKKKRKVED (contains the SV40 T antigen nuclear localization signal [*Kanayama et al., 2010*]). For this, the primer BglII-NLS-SalI-Fw (5'-TTT AGATCT ATG GCT AAA ATT CCT CCC AAA AAG AAA CGT AAA GTT GAA GAT TAA GTCGAC TTT-3') and BglII-NLS-SalI-Rev (5'-AAA GTCGAC TTA ATC TTC AAC TTT ACG TTT CTT TTT GGG AGG AAT TTT AGC CAT AGATCT AAA-3') (coding for the NLS sequence) were annealed to each other, digested with BglII and SalI and inserted to the already cut vector. This resulted in an EGFP-NLS-PolyA construct (see *Figure 5—figure supplement 1*).

The function of *Pt-Ets4* was analyzed either by injecting capped mRNA of the Pt-Ets4-EGFP fusion construct alone or by injecting capped mRNA of *Pt-Ets4* (the EGFP was removed from the EGFP-Pt-Ets4-PolyA construct via an XhoI digest) together with capped mRNA of EGFP-NLS. Ectopic expression of *Pt-Ets4* co-injected with EGFP-NLS resulted in the same phenotype as shown for the EGFP-Pt-Ets4 fusion construct.

To obtain a stronger signal during live imaging, capped mRNA of EGFP-Pt-Ets4 was co-injected with EGFP-NLS.

To mark the membranes of the embryonic cells and to visualize the delamination process of the Pt-Ets4 positive cell clones, capped mRNA coding for lynGFP (*Köster and Fraser, 2001*) was co-injected with capped mRNA coding for EGFP-Pt-Ets4. Next to the ectopic expression of lynGFP we tried to ectopically express GAP43YFP (another marker that was shown to localize to the cell membranes of *Tribolium* embryos [*Benton et al., 2013*, *Benton et al., 2016*]). However, we observed a much stronger signal for lynGFP.

## Whole mount *in situ* hybridization

Embryos used for RNA *in situ* hybridization were fixed in a two phase fixative containing 1.5 ml of PBS, 1.5 ml 10% formaldehyde and 3 ml heptane. Prior to fixation embryos were dechorionated for 3–5 min in a 2.8% hypochlorite solution (DanKlorix). The embryos were washed with $H_2O$ several times and were transferred to the fixative, subsequently. Embryos were fixed for over night at room temperature and 50–100 rpm. After the fixation embryos were gradually transferred (30%, 50%, 80%) to 100% methanol.

*In situ* hybridization was performed as previously described (*Prpic et al., 2008*) with minor modifications (proteinase K treatment was not carried out). Fluorescent FastRed staining was performed as described in (*Benton et al., 2016*).

### *In situ* hybridization on *Pt-Ets4* RNAi embryos

RNAi embryos were fixed at the desired stage. The development of several embryos of the same cocoon was monitored under oil. Only the embryos of severely affected cocoons were used for this analysis. For each round of *in situ* hybridization (as a control) the embryos from the same cocoons were analyzed for the expression of *Pt-Ets4*.

For the control and for the *Pt-Ets4* knockdown, more than 10 embryos were analyzed for the expression of each gene. Control and *Pt-Ets4* knockdown embryos were fixed according to the same fixation protocol (see above). Expression analysis of *Pt-Ets4*, *Pt-hb*, *Pt-fkh*, *Pt-dpp* and *Pt-fascin* in control and *Pt-Ets4* RNAi embryos (*Figure 2—figure supplement 1A and B*; *Figure 4*) was performed at the same time and stage (using the same solutions, probe concentrations and probes from the same probe synthesis reaction) and the color reaction was stopped at identical time points. As *Pt-fascin* is weakly activated by Pt-Ets4, the color reaction of the *Pt-fascin* staining in mid stage 5 embryos (*Figure 3D*) took much longer.

### Single embryo *in situ* hybridization

After the injection of capped mRNA coding for EGFP-Pt-Ets4, pictures of the single living embryos that exhibited an EGFP-Pt-Ets4 positive cell clone were taken. The same embryos were fixed by injecting 10% formaldehyde to the perivitelline space and were incubated for several hours at room temperature. To remove the oil, the single embryos were transferred to heptane and the chorion was removed using forceps. Subsequently, the single embryos were transferred to 100% methanol and the vitelline membrane was removed using forceps. *In situ* hybridization on the single embryos was performed as described above. The expression analysis of each gene was performed in three control (injection of capped mRNA coding for EGFP) and three EGFP-Pt-Ets4 overexpression embryos.

## pMAD antibody staining

To analyze BMP pathway activity in control and *Pt-Ets4* knockdown embryos a pMAD antibody staining was performed in embryos that were already stained for *Pt-otd* (via *in situ* hybridization, n > 10 for control and *Pt-Ets4* RNAi embryos). Embryos were fixed as described above.

*In situ* stained embryos were washed in PBST (3 × 15 min) and blocked in PBST containing 0,1% BSA and 5% goat serum (1 hr at RT). Subsequently, the embryos were transferred to a fresh solution of PBST containing 0,1% BSA and 5% goat serum. The Phospho-Smad1/5 (Ser463/465) (41D10) Rabbit mAb (Cell Signaling Technology, Inc. (RRID:AB_491015)) was added to this solution (antibody concentration: 1:1000; 4°C o.n.). On the next day the embryos were washed in PBST (3 × 15 min) and were blocked again in PBST containing 0,1% BSA and 5% goat serum (1 hr at RT). The secondary antibody (Anti-Rabbit IgG, couplet to alkaline phosphatase (AP), produced in goat; A3687 SIGMA (RRID:AB_258103)) was added to the blocking solution at a 1:1000 concentration. After incubating the secondary antibody for 2–3 hr (RT) excessive antibody was removed by washing the embryos several times in PBST (6 × 15 min; final washing step at 4°C o.n.). Finally, a regular NBT/BCIP staining was carried out (see whole mount *in situ* hybridization). For the control and the *Pt-Ets4* RNAi embryos, the staining reaction was stopped at identical time points

## EGFP antibody staining

We used an anti green fluorescent protein mouse IgG antibody (A11120; ThermoFischer SCIENTIFIC (RRID:AB_221568); final concentration 1:1000) as primary and an Alexa Fluor 488 goat anti mouse IgG (A11001; ThermoFischer SCIENTIFIC (RRID:AB_2534069); final concentration 1:400) as secondary antibody.

## Durcupan sections

To stain all of the embryonic cells, *in situ* hybridization with the ubiquitously expressed gene *Pt-arm* was carried out in early stage 5 embryos of control and *Pt-Ets4* RNAi embryos. Embryos were then stained with Sytox Green (1:5000 in PBST, ThermoFischer SCIENTIFIC). Embryos were then gradually (50%, 70%, 90%) transferred to 100% EtOH. After a washing step in 1:1 EtOH/acetone, the embryos were transferred to 100% acetone. Single embryos were transferred to microtome embedding molds in a 1:1 durcupan/acetone solution. Acetone was removed by incubating the embryos at

room temperature (o.n.). The embedding molds were filled with fresh durcupan (Fluka). Polymerisation of the durcupan was carried out at 65°C (16–20 hr). Cross sectioning (8 µm) was performed on a LEICA RM 2255 microtome (n = 3 for control and *Pt-Ets4* RNAi embryos).

## Bioinformatics

RNA from stage 1–3 embryos was extracted and sequenced as described in the 'Identification of *Pt-Ets4*' section above. These sequences were made available for us to download from the Cologne Centre for Genomics server, and FastQC (RRID:SCR_014583) (*Andrews, 2010*) was used for initial assessment of read quality. This was excellent (lower quartile Phred quality above 30 until the last base in the read, no residual adapter sequence noted) and as such no trimming was performed. Comparative expression analysis was performed by mapping reads to *Parasteatoda_tepidariorum* AUGUSTUS gene predictions (https://i5k.nal.usda.gov/Parasteatoda_tepidariorum) using RSEM 1.2.28 (*Li and Dewey, 2011*) and Bowtie 1.0.0 (RRID:SCR_005476) (*Langmead, 2010*) as packaged in the Trinity 2.2.0 module (RRID:SCR_013048) (-est_method RSEM—aln_method bowtie [*Grabherr et al., 2011*]). Cross sample normalization was performed using Trimmed Mean of M-values, and edgeR (RRID:SCR_012802) (*Robinson et al., 2010*) was run to determine differential expression with a dispersion ratio fixed at 0.1. Those differentially expressed genes with a p-value cut off for FDR of 0.001 and min abs(log2(a/b)) change of 2 were then chosen for annotation and further examination, with target gene results provided in *Figure 1—figure supplement 1*.

## Imaging and image processing

Pictures were taken using an Axio Zoom.V16 that was equipped with an AxioCam 506 color camera. Confocal imaging was performed on a LSM 700 (Zeiss). Live imaging was carried out on the Axio Zoom.V16, a Zeiss AxioImager.Z2 (equipped with an AxioCam MRm camera and a movable stage) and on a Leica CLSM SP8 (Imaging facility Biocenter Cologne).

Projections of image stacks were carried out using Helicon Focus (HeliconSoft (RRID:SCR_014462)) or Fiji (*Schindelin et al., 2012*) (RRID:SCR_002285).

All movies have been recorded at room temperature and images have been adjusted for brightness and contrast using Adobe Photoshop CS5 (RRID:SCR_014199).

For false-color overlays of *in situ* hybridization images a bright field image of the NBT/BCIP staining was inverted. This inverted picture was pasted into the red channel of the nuclear stain image. The input levels (Adobe Photoshop CS5; Levels function) of the red channel were adjusted in a way that only the signal of the NBT/BCIP staining remained visible.

## Acknowledgements

We thank Evelyn Schwager for details on the durcupan-sectioning protocol and Mette Handberg-Thorsager for providing the lynEGFP construct. We thank Hiroki Oda and Maarten Hilbrant for comments on the manuscript.

## Additional information

### Funding

| Funder | Grant reference number | Author |
| --- | --- | --- |
| Deutsche Forschungsgemeinschaft | PE 2075/1-1 and PE 2075/1-2 | Matthias Pechmann |
| Alexander von Humboldt-Stiftung | Fellowship for Postdoctoral Researchers | Matthew A Benton |
| Volkswagen Foundation | project number: 85 983 | Nico Posnien |
| Deutsche Forschungsgemeinschaft | PO 1648/3-1 | Nico Posnien |
| Deutsche Forschungsgemeinschaft | SFB680 | Siegfried Roth |

The funders had no role in study design, data collection and interpretation, or the decision to submit the work for publication.

## Author contributions

MP, Conceived the project, designed and performed the experiments, interpreted the results and wrote the initial draft of the paper; MAB, Designed and performed the experiments. Performed the phylogenetic analysis. Interpreted the results, contributed to the writing of the final manuscript and approved the final manuscript; NJK, NP, Performed the bioinformatic analyses. Contributed to the writing of the final manuscript and approved the final manuscript; SR, Designed the experiments. Interpreted the results. Contributed to the writing of the final manuscript and approved the final manuscript

## Author ORCIDs

Matthias Pechmann, http://orcid.org/0000-0002-0043-906X
Matthew A Benton, http://orcid.org/0000-0001-7953-0765
Nathan J Kenny, http://orcid.org/0000-0003-4816-4103
Nico Posnien, http://orcid.org/0000-0003-0700-5595
Siegfried Roth, http://orcid.org/0000-0001-5772-3558

## Additional files

### Supplementary files

• Supplementary file 1. Full sequence of the EGFP-Pt-Ets4-PolyA fusion construct

### Major datasets

The following dataset was generated:

| Author(s) | Year | Dataset title | Dataset URL | Database, license, and accessibility information |
|---|---|---|---|---|
| Pechmann M, Benton M, Posnien N, Kenny NJ, Roth S | 2017 | *Parasteatoda tepidariorum* transcriptomes of embryonic stages 1-3 | http://www.ncbi.nlm.nih.gov/bioproject/383558 | Publicly available at NCBI BioProject (accession no. PRJNA383558) |

The following previously published dataset was used:

| Author(s) | Year | Dataset title | Dataset URL | Database, license, and accessibility information |
|---|---|---|---|---|
| i5k initiative | 2014 | Augustus_gene_set-Primary_Gene_Set | https://i5k.nal.usda.gov/data/Arthropoda/partep-%28Parasteatoda_tepidariorum%29/Current%20Genome%20Assembly/2.Official%20or%20Primary%20Gene%20Set/Augustus_gene_set-Primary_Gene_Set/ | Publicly available at the United States Department of Agriculture National Agricultural Library (https://i5k.nal.usda.gov/Parasteatoda_tepidariorum) |

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
