## [Decision Letter]

Thank you for submitting your article "A novel role for *Ets4* in axis specification and cell migration in the spider *Parasteatoda tepidariorum*" for consideration by *eLife*. Your article has been reviewed by two peer reviewers, and the evaluation has been overseen by a Reviewing Editor and Diethard Tautz as the Senior Editor. The following individual involved in review of your submission has agreed to reveal his identity: Prashant P Sharma (Reviewer #3).

The reviewers have discussed the reviews with one another and the Reviewing Editor has drafted this decision to help you prepare a revised submission.

Summary: This work provides important insights into the maintenance of the cumulus in the spider, *Parasteatoda tepidariorum*. The cumulus serves as an early organizer in spider embryos, and this manuscript provides important insights into the control of cumulus migration by *Ets4*. The work also provides an excellent example of using various recent technologies to make rapid progress in new model systems. It also provides very important insights into the molecular control of an organizer and will allow for comparisons of organizer function between phylogenetically distant species.

Essential revisions:

1) The authors should describe the treatment of the control and RNAi embryos in more detail, i.e. did the authors do the fixation and *in situ* hybridisation of the control and RNAi embryos in parallel using the same solutions? Was the duration of the staining the same for control and RNAi embryos?

2) The authors should submit stage-specific transcriptomes to NCBI and including sample sizes for all experiments.

---

## [Author Response]

*Essential revisions:*

*1) The authors should describe the treatment of the control and RNAi embryos in more detail, i.e. did the authors do the fixation and in situ hybridisation of the control and RNAi embryos in parallel using the same solutions? Was the duration of the staining the same for control and RNAi embryos?*

These are important details and we have added the missing information to the manuscript.

As the injected spider females do not produce the cocoons at the same time, it is impossible to fix all of the analysed embryos in parallel. However, all embryos were fixed according to a well-established fixation protocol. We have added a detailed description of this fixation protocol to the Materials and methods section “Whole mount *in situ* hybridisation”.

Whenever we compared the expression of a given gene, we have performed RNA *in situ* hybridizations in parallel using the same solutions, probe concentrations and probes from the same probe synthesis reaction. For these experiments, the duration of the staining was also the same for control and RNAi embryos. Expression analysis of *Pt-Ets4, Pt-hb, Pt-fkh, Pt-dpp* and *Pt-fascin* (Figure 2—figure supplement 1; Figure 4) as well as the pMad antibody staining (Figure 3) in control and Pt-Ets4 RNAi embryos has been performed at the same time and the color reaction was stopped at identical time points. We have added this information to the Materials and methods sections “*In situ* hybridization on Pt-Ets4 RNAi embryos” and “pMad antibody staining”.

As *Pt-fascin* is weakly activated by Pt-Ets4, the color reaction of the *Pt-fascin* staining in mid stage 5 embryos (Figure 3) took much longer. We have added this information, too.

*2) The authors should submit stage-specific transcriptomes to NCBI and including sample sizes for all experiments.*

The stage-specific transcriptomes have been publicly released and are available under the BioProject ID: PRJNA383558. This information is also provided in the Materials and methods section “Identification and cloning of *Pt-Ets4* sequence”.

Where missing, we have added sample sizes for the experiments to the respective Materials and methods section.